# Tumor-Associated Fibroblast-Derived Exosomal circDennd1b Promotes Pituitary Adenoma Progression by Modulating the miR-145-5p/ONECUT2 Axis and Activating the MAPK Pathway

**DOI:** 10.3390/cancers15133375

**Published:** 2023-06-27

**Authors:** Qian Jiang, Zhuowei Lei, Zihan Wang, Quanji Wang, Zhuo Zhang, Xiaojin Liu, Biao Xing, Sihan Li, Xiang Guo, Yanchao Liu, Xingbo Li, Yiwei Qi, Kai Shu, Huaqiu Zhang, Yimin Huang, Ting Lei

**Affiliations:** 1Sino-German Neuro-Oncology Molecular Laboratory, Department of Neurosurgery, Tongji Hospital of Tongji Medical College, Huazhong University of Science and Technology, Jiefang Avenue. 1095, Wuhan 430030, China; m202076460@hust.edu.cn (Q.J.);; 2Hubei Key Laboratory of Neural Injury and Functional Reconstruction, Huazhong University of Science and Technology, Wuhan 430030, China; 3Department of Orthopedics, Tongji Hospital of Tongji Medical College of Huazhong University of Science and Technology, Jiefang Avenue. 1095, Wuhan 430030, China

**Keywords:** pituitary adenoma, exosome, tumor-associated fibroblasts, circDennd1b, miR-145-5p, ONECUT2, FGFR3

## Abstract

**Simple Summary:**

The tumor microenvironment, especially tumor-associated fibroblasts (TAFs), has been extensively studied in cancer, but not as much in pituitary adenoma (PA). Studies of TAFs will provide a better understanding of the mechanisms by which PA exhibits aggressive behavior. Our study proved that TAFs promote PA progression through exosomal circDennd1b. circDennd1b, as a ceRNA, upregulated the expression of the target gene ONECUT2 by sponging miR-145-5p, thereby transcriptionally regulating FGFR3 and activating the downstream MAPK pathway and finally promoting the PA progression. Moreover, the suppression of ONECUT2 and TAFs can improve the efficacy of clinical drugs for PA.

**Abstract:**

TAF participated in the progression of various cancers, including PA via the release of soluble factors. Exosomes belonged to extracellular vesicles, which were revealed as a crucial participator in intercellular communication. However, the expression pattern and effect of TAF-derived exosomes remained largely unknown in PA. In the present study, we performed in silico analysis based on public RNA-seq datasets to generate the circRNA/miRNA regulatory network. The qRT-PCR, Western blotting, RNA pull-down, and luciferase assay were performed to investigate the effect of TAF-derived exosomes. TAF-derived exosomal circDennd1b was significantly upregulated in PA and promoted the proliferation, migration, and invasion of PA cells via sponging miR-145-5p in PA cells. In addition, miR-145-5p directly regulated One Cut homeobox 2 (ONECUT2/OC2) expression and inhibited the promoting effect of ONECUT2 on PA. We further demonstrated that ONECUT2 transcriptionally increased fibroblast growth factor receptor 3 (FGFR3) expression, which further activates the mitogen-activated protein kinases (MAPK) pathway, thus promoting PA progression. Moreover, the suppression of TAFs by ABT-263 and ONECUT2 by CSRM617 inhibited the growth of PA. In conclusion, our study illustrated that TAF-derived exosomal circDennd1b affected PA progression via regulating ONECUT2 expression, which provides a potential therapeutic strategy against aggressive PA.

## 1. Introduction

PA accounted for approximately 17% of all intracranial tumors [1]. Histologically, PA is a benign tumor, and almost 35% of PA displays an aggressive pattern of growth. Others and our clinical observations revealed that the invasive and proliferous nature of PA often led to adverse outcomes such as difficulty in surgical treatment, the high probability of recurrence, repeated surgeries, and poor prognosis, which pose a severe threat to the physical and mental health of patients [2,3,4,5,6]. Research into the mechanisms of PA aggressiveness might help to treat PA and improve patients’ quality of life.

Various in intracellular signaling have been previously identified by our group and others as crucial factors promoting PA aggressive behavior, such as a disintegrin and metalloprotease 12 (ADAM12)/epidermal growth factor receptor (EGFR) [7], growth hormone secretagogue receptor 1a (GHSR-1a) [8], and general transcription factor IIB (GTF2B) [9] signaling. Although there have been many clinical cohort studies and basic research on PA, it was still necessary to explore the mechanisms in depth. In recent years, accumulated evidence has demonstrated that the tumor microenvironment, which interacts closely with the tumor cells, has attracted attention and is considered as an essential force driving tumor progression. Tough textured tumors predicted a higher probability of invasion and recurrence [10,11,12]. Tumor-associated stromal cells can be recruited and transformed into a pro-tumor phenotype, assisting the proliferation and invasion of tumor cells. Compared to resting fibroblasts, TAFs have enhanced proliferative and migratory properties and exhibit differential gene and protein expression patterns, promoting tumor progression through various pathways [13,14,15]. The specific role of TAFs in the proliferation and invasion of PA remained unclear, which warranted further investigation.

Exosomes are extracellular vesicles with a diameter range of 40–160 nm, surrounded by a lipid bilayer, and secreted by most eukaryotic cells, which can be delivered to target cells [16]. Exosomes and their cargoes can be used as prognostic markers and therapeutic targets for cancer, and even anticancer drug carriers [17]. Regarding PA, it has also been suggested that exosomes can act as carriers to transport non-coding RNA to act on tumor cells [18]. CircRNAs competitively bind to target microRNAs to promote or inhibit the function of the corresponding microRNAs and play a vital role in tumorigenesis by affecting the invasive and migratory capacity of cancer cells [19]. This competitive binding of mRNAs is known as miRNA sponge action [20]. However, whether circRNAs bridging TAFs to mediate PA progression is still unknown.

In the present study, we aimed to investigate the role of TAF-derived exosomal circRNA on PA growth and its underlying mechanisms. Moreover, the potential treatment efficiency of CSRM617 or ABT-263 in PA was also considered. Our findings might provide novel thoughts in PA progression from a cell–cell interaction perspective and thus offer potential targets for the treatment of aggressive PA.

## 2. Materials and Methods

### 2.1. Data Extraction and Analysis

CircRNA annotations and sequences were extracted from circbase. MiRNA sequences were extracted from starbase starbase.sysu.edu.cn, accessed on 30 August 2022). The conserved circRNA sequences were extracted from circAtlas (http://circatlas.biols.ac.cn, accessed on 18 June 2022). circRNA sequence was used by Qiu Du [21], etc. Circinteractome (circinteractome.nia.nih.gov, accessed on 30 January 2022) was used to analyze the potential binding of miRNAs to individual circRNA. The potential targets of individual miRNAs were predicted by starbase. miRNA microarray dataset (GSE46294) and mRNA microarray datasets (GSE169498 and GSE51618) were downloaded from NCBI GEO public database (www.ncbi.nlm.nih.gov, accessed on 29 January 2022) and analyzed by R version 3.4.3. The circRNA–miRNA was visualized by Cytoscape (version 3.9.0). The mRNA chip seq dataset (CistromeDB: 104814) was downloaded by Cistrome Data Browser (cistrome.org/db, accessed on 25 February 2022). The proliferation and migration datasets were downloaded from GSEA.

### 2.2. Cell Culture and Transfection

The rat PA cell lines (GH3, MMQ) and mouse PA cell lines (TtT/GF, AtT20) were provided by Dr. Nic Savaskan, Erlangen University Hospital. Briefly, GH3, TtT/GF, and AtT20 cell lines were routinely maintained in DMEM supplemented with 10% FBS and penicillin/streptomycin (10 mg/mL solution diluted 1:100) in a humidified atmosphere of 5% CO_2_ at 37 °C. MMQ cell lines were maintained in 1640 medium, others being equal.

For cell transfection, the GV657 vector for overexpressing ONECUT2 in rat was purchased from Genechem (Shanghai, China), GV657 vector for overexpressing ONECUT2 in mouse was purchased from Tsingke (Beijing, China), and PLVX-puro circRNA vector for overexpressing circDennd1b in rat, and TK-PCDH-copGFP-T2A-Puro circRNA vector for overexpressing circDennd1b in mouse. The detailed sequence was available in Appendix A. miR-145-5p mimic, inhibitor, agomir, and antagomir, and their control plasmids (NC mimic, NC inhibitor, NC agomir, and NC antagomir, respectively) were purchased from Tsingke (Beijing, China). Cells were transfected with Lipofectamine 3000 (Invitrogen, Carlsbad, CA, USA).

Specifically, for the FGF-treated group, FGF recombinant protein was purchased from MCE (cat: HY-P7179) and added to GH3 or AtT20 cells at a concentration of 100 ng/mL, and the proliferation and migration of cells was observed after 2 days.

### 2.3. PA Tissues Collection

The PA tissues were from Tongji Hospital, affiliated with Tongji Medical College. A total of 49 pairs of PA organizations were included. All human-related procedures implemented in our study were under the 1964 Declaration of Helsinki and approved by the Ethics Committee of Tongji Hospital Affiliated to Tongji Medical College (TJ-IRB20220325). The informed consent of all participants should be obtained before proceeding with any study-related procedures. All samples were immediately snap-frozen in liquid nitrogen and stored at −80 °C.

### 2.4. Isolation of Primary PA Cells and Fibroblasts

For primary PA cells, PA tissues were minced and digested in trypsin (cat: G4001, Servicebio, Wuhan, China) for 30 min and then cultured in DMEM containing 10% fetal bovine serum for subsequent experiments.

For primary fibroblasts, tumors and skin tissues were minced and digested in type I collagenase (cat: 40507ES60, Yeasen, Shanghai, China) and then cultured in DMEM containing 10% fetal bovine serum at 37 °C until fibroblasts attached to the dish. Anti-Fibroblast MicroBeads (cat: no. 130-050-601 and no. 130-116-474, Miltenyi Biotec, Bergisch Gladbach, Germany) and anti-fibroblast-activating protein (cat: ab218164; Abcam, Cambridge, UK) were prepared to isolate TAFs and NFs. The cells were incubated with FAP antibodies in ice for 1 h. After washing, microbead-coupled secondary antibodies were added, and FAP+ cells were separated into magnetic columns after rinsing. Primary fibroblasts were used prior to passage 5.

Primary fibroblasts of rats and mice were derived from skin tissues, and the extraction method was consistent with that of human primary fibroblasts. For primary TAF cells of rats and mice, tumor cells were mixed with primary fibroblasts for tumor grafting in the ratio of 3:1. Tumor tissues were digested after the tumor was removed, and the extraction method was consistent with that of human primary fibroblasts.

### 2.5. Isolation and Identification of Exosomes from the Cell Culture Medium

After incubation in the conditioned medium for 24 h, the medium containing 10% exosome-free FBS was centrifuged at 300× *g* and 2000× *g* to discard cell debris. The supernatant was then centrifuged at 10,000× *g* for 30 min to remove large-size shedding blisters. Finally, the supernatant was hypercentrifuged at 100,000× *g* for 90 min (OPTIMA XPN-100) with exosomes in the precipitate and resuspended in 1× PBS, and filtered through a 0.2 μm filter. All steps were performed at 4 °C. The isolated exosomes were identified by Chi Biotechnology (Shenzhen, China), including transmission electron microscopy (TEM) and nanoparticle tracking analysis (NTA).

### 2.6. qRT-PCR

Total RNA was isolated from cultured cells and tissues using Trizol reagent (Invitrogen, Carlsbad, CA, USA). The extracted RNA was then reverse-transcribed into cDNA by using Hifair^®^ III 1st Strand cDNA Synthesis SuperMix for qRT-PCR (gDNA digester plus) (cat: 11141ES60) (Yeasen, Shanghai, China). In addition, for miRNA and circRNA, Hifair^®^ III 1st Strand cDNA Synthesis SuperMix for qRT-PCR (gDNA digester plus) (CAT: 11139ES60) was used to perform reverse transcription. The primers used in the current experiments were designed and purchased from Tsingke Biotechnology; the primer sequence was listed in the Appendix A, and qRT-PCR was carried out on an ABI QuantStudio Real-Time PCR System (Thermo Fisher, Waltham, MA, USA). GAPDH, U6, and β-actin were applied as internal references for mRNAs, miRNAs, and circRNAs, respectively. The detail of the experiment followed Yeasen’s PCR kit. The relative expression levels were calculated by the 2^−ΔΔCT^ method.

### 2.7. Cell Viability Experiments

CCK-8 assay (Servicebio, Wuhan, China) tested cell viability in different groups according to the manufacturer’s protocol. Wells with a density of 5000 cells were seeded in 96-well plates. After 48 h, 10 μL of CCK8 reagent was added to each well with TtT/GF cells for 24 h. After incubation, the absorbance of the cells was captured and recorded at 490 nm using Infinite^®^ F50 (Tecan, Männedorf, Switzerland).

### 2.8. Colony Formation Assay

Different groups of transfected PA cells were seeded into six-well plates. After 2 weeks, TtT/GF cells needed 5 days after inoculation, cell colonies were fixed with 10% formaldehyde for 30 min, followed by 0.5% crystal staining for 30 min. Mobile phones were used to photograph the colonies.

### 2.9. Wound Healing Assay

Different groups of transfected PA cells were seeded in six-well plates and grown until ~90% confluence. The cell monolayer was wounded by scratching with a 200 µL pipette tip. The cell movement was captured by a microscope (Olympus, Tokyo, Japan) at 0 and 24 h, respectively.

### 2.10. Transwell Assay

Transwell chambers (Corning, NY, USA) were used to perform migration and invasion assay. First, 200 μL of serum-free medium mixed with 20,000 AtT20 or 10,000 TtT/GF cells was placed in the upper chamber; when performing an invasion experiment, Matrigel (Corning, NY, USA) was applied in the chamber beforehand. Meanwhile, 700 µL culture medium with 10% FBS was added into the lower chamber. For conditioned medium, instead, TAF conditioned medium (TAF-CM) or NF conditioned medium (NF-CM) was added in the lower chamber. The migrated or invaded cells were fixed with 4% paraformaldehyde for 30 min and stained with 0.5% crystal violet for 30 min, with proportions calculated under a microscope after 24 h.

### 2.11. Immunofluorescence (IF)

IF staining was performed using PA cells in 12-well plates. Subsequently, 5% BSA (Servicebio, Wuhan, China) was used for blocking, and then primary antibody was added: anti-β-tubulin (Proteintech, Wuhan, China, cat: 10094-1-AP, 1:200) at 4 °C overnight. Next, cells were incubated with a Cy3-conjugated anti-rabbit secondary antibody (Proteintech, Wuhan, China, cat: SA00009-3, 1:200) mixed with diamidino-2-phenylindole for 2 h. Finally, cells were stained with DAPI (Servicebio, cat: G1012) for 15 min. Data were acquired and analyzed thereafter.

For IF of tissues, after dewaxing, antigen repair, and blocking, the tissues were incubated with ki-67 (Proteintech, Wuhan, China, cat: 27309-1-AP) or MMP9 (Proteintech, Wuhan, China, cat: CL594-10375) antibody overnight, washed three times with PBST. Next, cells were incubated with a Cy3-conjugated anti-rabbit secondary antibody (Proteintech, Wuhan, China, cat: SA00009-3, 1:200) mixed with diamidino-2-phenylindole for 2 h. Finally, tissues were stained with DAPI (Servicebio, cat: G1012) for 15 min.

### 2.12. Immunohistochemistry (IHC)

Firstly, paraffin-embedded tissues were heated in a microwave oven at 60 °C for 2 h for dewaxing, then rehydrated by gradient alcohol and xylene. Then, the EDTA repair solution (cat: G1203, Servicebio, Wuhan, China) was used for antigen repair, following the manual. The tissues were blocked by 5% BSA for 1 h; then, the primary antibodies—α-SMA (Santa Cruz, CA, USA, cat: sc-32251), dilution ratio 1:200; PDGFRB (Proteintech, Wuhan, China), dilution ratio 1:50; TAGLN (Proteintech, Wuhan, China), dilution ratio 1:20, FGFR3 (Proteintech, Wuhan, China, cat: 66954-1-Ig), dilution ratio: 1:200—were applied overnight at 4 °C. After 3% H_2_O_2_ blocking endogenous catalase, tissues were followed by the anti-rat or anti-rabbit secondary antibody incubation for 2 h, then SABC was used at 37 °C incubation for 30 min. After 1:50 dilution of DAB staining and nuclear staining, the tissues were used for subsequent photography. ImageJ was applied to analyze the IHC staining. Specific method was followed: open → file → Image → Type → RGB stack → turned scroll bars to blue → Image → Adjust → Threshold → auto → set → Analyze → Set Measurements.

### 2.13. Western Blot Assay

Different groups of GH3 and AtT20 cells were lysed on ice with RIPA buffer. The isolated exosomes can be directly mixed with the loading buffer. Then, protein lysates were electrophoresed on 10% SDS polyacrylamide gels, transferred onto PVDF membranes (Millipore, Burlington, MA, USA), and blocked with NcmBlot blocking buffer (NCM Biotech, Suzhou, China). Afterward, primary antibodies: anti-ONECUT2 (Biorbyt, Cambridge, UK, cat: orb49419, 1:1000), anti-PI3K (Proteintech, Wuhan, China, cat: 60225-1-Ig, 1:1000), anti-P-PI3K (Thermofisher, Waltham, MA, USA, cat: PA5-104853, 1:1000), anti-PKCδ (Proteintech, Wuhan, China, cat: 14188-1-AP, 1:1000), anti-P-PKCδ (Proteintech, Wuhan, China, cat: 29562-1-AP, 1:1000), anti-mTOR (Protentech, Wuhan, China, cat: 66888-1-Ig, 1:1000), anti-P-mTOR (Proteintech, cat:67778-1-Ig, 1:1000), anti-AKT (Proteintech, cat: 60203-2-Ig, 1:1000), anti-P-AKT (Proteintech, Wuhan, China, Cat: 80455-1-RR, 1:1000), anti-ERK (Proteintech, Wuhan, China, cat: 11257-1-AP, 1:1000), anti-P-ERK (Proteintech, Wuhan, China, cat: 80031-1-RR, 1:1000), anti-GAPDH (Proteintech, Wuhan, China, Cat: 60004-1-Ig, 1:1000), anti-CD63 (Santa Cruz, CA, USA, cat: sc-5275;1:200), anti-CD9 (Santa Cruz, CA, USA, cat: sc-13118; 1:200) were added at 4 °C overnight. Subsequently, secondary antibodies (1:5000; Proteintech, Wuhan, China) were incubated for 2 h.

### 2.14. Cell Cycle Assay

PA cells were cultured in 6-well plates at a concentration of 2 × 10^5^ cells per well. After transfection for 48 h, PA cells were filtered overnight at −20 °C in 75% ethanol and then stained with PI/RNase staining buffer (cat: 40301ES50, Yeasen, Shanghai, China). Cell cycle distribution was measured by flow cytometry, and cell cycle output was further analyzed using ModFit 5.0.

### 2.15. PKH26 Staining

The PKH26 Red Fluorescent Cell Link Kit (Sigma, Osterode, Germany) was used for exosome staining. Exosomes resuspended in 1000 μL of diluent C were mixed with 2 μL of PKH26 dye solution and held for 5 min, then stopped by adding 1% BSA. Exosomes were centrifuged at 100,000× *g* and the precipitate was used for subsequent operations.

### 2.16. RNA Electrophoretic

Divergent primers and convergent primers of circDennd1b were designed by Servicebio (Wuhan, China), and the experiment was divided into control and RNase treatment groups after amplification. A 1% agarose gel with 1× TAE was applied to perform electrophoresis for 20 min. Results were photographed with a gel imager (BIO-RAD GelDoc XR, Hercules, CA, USA).

### 2.17. 3D Invasion Assay

The Ultra-Low Cluster 96-well plate (Corning, NY, USA) was applied with 180 μL of complete medium mixed with 5000 TtT/GF cells and 20 μL of collagen (cat: 40125ES10, Yeasen, Shanghai, China) per well, then the 96-well plate was centrifuged at 4 °C and 1000× *g* for 10 min to form the cell sphere. Photographs were taken after 1 day and 5 days.

### 2.18. CircRNA Pull-Down

Biotin-labeled circDennd1b probe and control probe (Genechem, Shanghai, China) were used for circRNA pull-down, and the assay was performed as mentioned previously. In brief, 293T and AtT20 were crosslinked by 1% formaldehyde for 30 min, then lysed in co-IP buffer, and centrifuged. The supernatant was incubated with circDennd1b-specific probes–streptavidin beads (Life Technologies, New York, NY, USA) mixture overnight at 37 °C. On the next day, the samples were washed and incubated with lysis buffer and proteinase K. Finally, the mixture was added with Trizol reagent for RNA extraction and followed by detection of miR-145-5p and U6.

### 2.19. Dual-Luciferase Reporter Assay

293T cells were cultured and seeded into 24-well plates. Then, the GV272 reporter vector (Genechem, Shanghai, China) carrying wild-type (WT) or mutant type (MUT) circDennd1b and ONECUT2 were transfected into 293T cells in combination with miR-145-5p mimic or NC mimic by Lipofectamine 3000 (Invitrogen, Carlsbad, CA, USA). GH3 or AtT20 was used to exam dual luciferase in rats or mice. For experimental procedures, refer to the dual luciferase reporter gene assay kit (cat: 11402ES60, Yeasen, Shanghai, China).

For promoter truncation experiments, JASPAR online tool was used to analyze (ONECUT2: MA0756.2, FGFR3: NC_000004.12) the binding sites. Two possible binding sites were identified, which were located at sites 10–18 and 660–668 in the promoter region of FGFR3. We then designed four promoter sequences and transfected them into GV272 vector (Genechem, Shanghai, China). For experimental procedures, refer to the dual luciferase reporter gene assay kit (cat: 11402ES60, Yeasen, Shanghai, China).

### 2.20. In Vivo Xenograft Assay

BALB/c nude mice (6 weeks old) were housed at Tongji Medical College. The animal research was approved by the Animal Research Ethics Committee of Tongji Medical College, Huazhong University of Science and Technology (TJH-202206015). The mice were randomly assigned to two or four groups for further study. A total of 1 × 10^7^ cells, including transfected and control PA cells, were subcutaneously injected into each mouse. For the group injected with miR-145-5p antagomir, nude mice were injected four time with 5 nmol every three days. For the group injected with miR-145-5p agomir, nude mice were injected four times with 2 nmol every three days. For the tumor cells and fibroblast mixed tumor group, 7.5 × 10^6^ PA cells were mixed with 2.5 × 10^6^ NFs or TAFs. For the drug treatment group, OCT was administered at 30 μg/kg, and bromocriptine at 500 μg/kg, CSRM617 (MCE, cat: HY-122611A, South Brunswick, NJ, USA) was administered at a dose of 50 mg/kg via oral administration every day, ABT-263 (MCE, cat: HY-10087, South Brunswick, NJ, USA) was administered at a dose of 75 mg/kg via intraperitoneal injection in a total volume of 100 μL every second day. Tumor size and weight were measured at two weeks after injection. Tumor size was calculated as: 0.5 × width^2^ × length. Tumor samples were stored by paraformaldehyde fixation and paraffin-embedding for further experiments.

### 2.21. Masson Staining

Paraffin-embedded tissues were heated in a microwave oven at 60 °C for 2 h for dewaxing, then rehydrated by gradient alcohol and xylene. Next, the potassium dichromate staining was performed for 30 min, followed by nuclear staining for 5 min. Then, the hydrochloric acid alcohol was used for differentiation, and ammonia was applied to stain the nucleus blue. Finally, the bright green dye solution was used to stain collagen fibers for 30 min.

### 2.22. Tunel Staining

Paraffin-embedded tissues were heated in a microwave oven at 60 °C for 2 h for dewaxing, then rehydrated by gradient alcohol and xylene. Then, the EDTA repair solution (cat: G1203, Servicebio, Wuhan, China) was used for antigen repair, following the manual. The tissues were blocked by 5% BSA for 1 h. Then, the tissues were incubated overnight with primary antibody anti-alpha-smooth muscle actin (α-SMA) (Santa Cruz, CA, USA, cat: sc-32251). After washing three times by PBST, tissues were incubated with Cy3-conjugated anti-rabbit secondary antibody (Proteintech, Wuhan, China, cat: SA00009-3, 1:200), followed by apoptosis staining, which was performed according to the apoptotic kit (cat: A112-01: 20 rxn, Vazyme, Nanjing, China) and finally incubated with DAPI for 15 min. A microscope captured the results.

### 2.23. Statistical Analysis

GraphPad Prism (version 9.3) and R version (4.2.0) were used for statistical analyses. All data for this study were presented as mean ± standard deviation. Pearson correlation and linear regression were used to compare these parameters between two variables. One-way ANOVA was used for multi-sample comparison and Student’s *t*-test was used for comparison between two groups. Statistical significance was considered to be indicated by a value of *p* < 0.05.

## 3. Results

### 3.1. TAFs Promoted the Proliferation and Migration of PA Cells

TAFs have been proved to play a significant role in tumor progression [13,14,22,23]. To evaluate the relationship between TAFs and PA growth, we collected information on 242 PA patients in the department of neurosurgery of Tongji Hospital from January 2015 to January 2022, revealing that tough-textured tumors had larger tumor volume and were more invasive than soft-textured tumors, with the Knosp grade used as an index to measure PA invasiveness: Knosp 0 suggested no invasion and higher Knosp grade meant more invasion (Figure 1a). In addition, males seemed to have a higher proportion of tough-textured tumors (Table 1). Moreover, 49 specimens of patients with PA were collected. IHC staining for alpha-smooth muscle actin (α-SMA), platelet-derived growth factor receptors beta (PDGFRB), and transgelin (TAGLN) was used to assess TAFs density. The results suggested that TAFs density was higher in PA with tough texture (Figure 1b,c). Moreover, TAFs density was positively correlated with tumor volume (Figure 1d). TAFs density was higher in Knosp 3–4 (Figure 1e). Previous studies have indicated that TAFs could secrete factors to promote pituitary tumor progression [24]. To further investigate the regulatory effect of TAF-derived factors on PA cells, TAFs from the tumor and normal fibroblasts (NFs) from the skin were isolated using magnetic activated cell sorting (Figure 1f). Conditioned medium (CM) of TAFs and NFs was harvested to treat PA cells. GH3, AtT20, and two PA primary cells exhibit higher cell viability in the presence of TAF-CM compared to NF-CM (Figure 1g), and AtT20 displayed more vigorous migration after treated with TAF-CM determined by Transwell assay (Figure 1h). Moreover, to verify the tumor promoting effect of TAFs in vivo, TAFs were mixed with PA cells and subcutaneously implanted into nude mice. Compared to the NFs-PA co-injection group, tumors with TAFs co-injection exhibit notably larger volume, indicating the tumor-promoting effect of TAFs (Figure 1i).

To sum up, we identified that TAFs density correlated with tumor volume and aggressiveness, and that TAF-CM could promote the proliferation and migration of PA cells.

**Table 1 cancers-15-03375-t001:** Correlation analysis between tumor texture and basic information.

Factors	Soft Texture	Tough Texture	*p*
Gender (male/female)	84/96	39/23	0.039 *
Age	46.11 ± 13.47	46.58 ± 13.58	0.814

* Statistical significance was considered to be indicated by a value of *p* < 0.05, * *p* < 0.05.

### 3.2. TAF-CM Promoted PA Progression by Inhibiting miR-145-5p

The competitive endogenous RNA (ceRNA) was defined as some transcripts sequestered specific miRNAs and diminished their repressive effects on other transcripts [25]. There were studies suggested that miRNAs regulated tumor progression [26]. Therefore, we inferred that TAF-CM promoted PA progression through ceRNA. A ceRNA network composed of circRNAs and miRNAs was constructed based on PA RNA sequencing data (GSE46294) and published circRNA-seq data [21], which contains 501 conserved upregulated circRNAs and 18 downregulated miRNAs (Figure 1j, Appendix A). The miRNA expression in the normal pituitary has been normalized to 1 and the miRNA expression was presented as fold change. Eighteen of these miRNAs that were downregulated in PA were predicted to sponge circRNAs (Figure 1k). Then, the top five most notable downregulated miRNAs (according to the fold change) were chosen because the target miRNA should also be associated with invasiveness. The five miRNAs were selected and their expression levels in 15 PA tissues were detected by qRT-PCR. There was the most obvious negative correlation between miR-145 and TAFs content (Figure 1l). Besides, miR-145-5p has been proved to be related to tumor growth and invasion in many studies [27,28,29], and miR-145-5p was chosen as the target miRNA via analyzing circRNA–miRNA interaction using circinteractome (circinteractome.nia.nih.gov, accessed on 30 January 2022) and miRbase (www.mirbase.org, accessed on 13 February 2022). In addition, a significant decrease in miR-145-5p expression was revealed in PA cells after TAF-CM treatment (Figure 1m). To examine the effect of miR-145-5p in TAF-mediated PA growth, miR-145-5p mimic was transfected to the PA cells in the presence of TAF-CM. Indeed, miR-145-5p treatment rescued the TAF-CM induced enhanced proliferation (Appendix A and Figure 1n,o). In addition, in vivo experiments demonstrated the tumor suppressive effect of miR-145-5p on PA (Figure 1p). Taken together, our results suggested that TAF-CM might release some unknown “factors” to promote PA progression by suppressing miR-145-5p.

### 3.3. TAF-Derived circDennd1b Promoted the Proliferation, Migration, and Invasion of PA Cells

It was previously reported that fibroblasts could secrete exosomes to regulate the progression of tumor cells [30]. Thus, we next determined the participation of circRNA-carrying exosomes in TAFs-mediated PA growth by suppressing miR-145-5p as we identified above. Exosomes were first harvested and identified from TAFs and NFs by WB and TEM, the size of fibroblasts was shown by NTA at around 100 nm (Figure 2a). Next, 11 circRNAs containing multiple miR-145-5p binding sites were screened through circinteractome. Hsa_circ_0006324 was identified as the target circRNA, as it was most significantly elevated in PA tissues as determined by qRT-PCR (Figure 2b and Appendix A). Hsa_circ_0006324 was proved as a highly conserved circRNA via circAltlas (circatlas.biols.ac.cn, accessed on 18 June 2022), which also can be named as circDennd1b among different species. The circular form of Dennd1b was then verified by nucleic acid electrophoresis, and the cDNA samples were detected by divergent primer and convergent primer (Appendix A) after RNase R digestion. Our data indicated that circDennd1b was resistant to RNase R digestion, suggesting its circular structure (Figure 2c). In addition, TAF-derived exosomes contained higher expression level of circDennd1b than NF-derived exosomes (Figure 2d). Besides, exosomes from aggressive PA tissues had higher expression levels of circDennd1b than those from non-aggressive PA (Figure 2d). Next, the engulfment/absorbance of TAF-derived exosomes by PA cells were detected (Figure 2e), and the circDennd1b expression level was significantly increased in PA cells after stimulation with TAF-derived exosomes (Figure 2f).

To demonstrate the impact of circDennd1b on tumor progression, expression level of circDennd1b was examined in PA specimens from patients, and as expected, circDennd1b RNA level positively correlated with tumor size and aggressiveness (Appendix A). To further investigate the potential tumor promoting role of circDennd1b, we constructed PA cell lines overexpressing circDennd1b (Appendix A). The cell viability assay (Figure 2g), colony formation assay (Figure 2h), and cell cycle assay (Figure 2k) were conducted, and we observed that circDennd1b increased the proliferation of PA cells. Results from Transwell assay (Figure 2i), wound healing assay (Figure 2j), and 3D invasion assay (Figure 2l) proved that circDennd1b enhanced the migration and invasion of PA cells. Moreover, circDennd1b overexpression resulted in larger tumors in vivo (Figure 2m). Taken together, TAF-derived exosomal circDennd1b promoted the proliferation, migration, and invasion of PA.

### 3.4. Tumor Promoting Effect of circDenn1b Was Achieved by Sponging miR-145-5p in PA Cells

As mentioned earlier, miR-145-5p was suggested to be a target of circDennd1b. To further investigate the interaction between circDennd1b and miR-145-5p, dual-luciferase reporter assay was used to verify specific binding of circDennd1b with miR-145-5p (Figure 3a). RNA pull-down assays also showed direct binding between circDennd1b and miR-145-5p (Figure 3b). In addition, the competitive expression between miR-145-5p and circDennd1b in PA cells was detected when we transfected miR-145-5p mimic (Appendix A). Next, miR-145-5p reversed the promoting effect of circDennd1b in PA cells as assessed by the cell viability assay (Figure 3c), colony formation assay (Figure 3d), Transwell assay (Figure 3e), and wound healing assay (Figure 3f). In vivo experiments demonstrated that miR-145-5p agomir attenuated tumor growth in circDennd1b overexpressed PA cells (Figure 3g). Taken together, circDennd1b promoted the progression of PA cells by sponging miR-145-5p.

### 3.5. miR-145-5p Inhibited PA Progression by Suppressing ONECUT2 Expression

To further identify the downstream target genes of miR-145-5p, we performed in silico analysis on two PA RNA-seq datasets (GSE51618 and GSE169498) in combination with prediction targets of miR-145-5p using starbase (starbase.sysu.edu.cn, accessed on 30 August 2022). All 13 potential target genes regulated by miR-145-5p are labeled in the volcano plot (Figure 4a). ONECUT2 was the most significantly upregulated among them in invasive PA compared to non-invasive PA using RNA-seq datasets from GSE169498. Therefore ONECUT2 was selected as the target gene of miR-145-5p in PA. In highly invasive PA tissues, ONECUT2 expression level was significantly higher by immunohistochemistry (Figure 4b). The dual luciferase assay indicated the direct binding of miR-145-5p to ONECUT2 mRNA 8′ mer (Figure 4c). Furthermore, ONECUT2 mRNA expression was significantly elevated by circDennd1b overexpression in PA cells determined by qRT-PCR (Figure 4d). To further verify this, we constructed fibroblasts overexpressing circDennd1b and collected exosomes to treat the PA cells. As expected, circDennd1b and ONECUT2 expression levels were significantly elevated in PA cells, while miR-145-5p expression was suppressed in the presence of exosomes from circDennd1b overexpressed NFs (Figure 4e).

Next, we constructed PA cells overexpressing ONECUT2 and treated with miR-145-5p mimic (Figure 4f,g). Tumor promoting effect of ONECUT2 on PA cells was characterized by a serious of in vivo and in vitro experiments and miR-145-5p can notably reverse these promoting effects mediated by ONECUT2 overexpression (Figure 4h–k). Consistently, PA cells with ONECUT2 knockdown exhibited decreased proliferation and migration activity (Appendix A). Taken together, circDennd1b exerted a tumor promoting effect by sponging miR-145-5p and thus upregulated ONECUT2 in PA cells.

### 3.6. ONECUT2 Bound FGFR3 and Activated Downstream MAPK Pathways to Promote PA Progression

ONECUT2 was characterized as a transcription factor that participates in cancer development [31,32,33]. To further explore the mechanism that ONECUT2 regulated tumor progression, we analyzed the ChIP data of ONECUT2 from CistromeDB (http://cistrome.org/db, accessed on 25 February 2022) accession number: 104814) (GSM3136847) and found that ONECUT2-regulated genes were significantly enriched in the MAPK pathway. Subsequently, the expression of FGFR3 was the most elevated signatures in genes of MAPK signaling pathway as shown by volcano plot (Figure 5a,b). By IHC, the FGFR3 expression level was higher in the invasive PA group. In addition, the FGFR3 expression was positively correlated with the ONECUT2 expression level in patients’ tissues (Appendix A). Next, we validated that ONECUT2 knockdown led to decrease in the expression of FGFR3 (Figure 5c). Then, the binding of ONECUT2 to FGFR3 was detected by promoter truncation experiments. We identified two potential binding sites in the 3000 bp promoter region of FGFR3 that might be bound by ONECUT2 predicted by JASPAR (jaspar.genereg.net, accessed on 28 September 2022), where located at sites 10–18 and 660-668 in the promoter region of FGFR3. Subsequently, we designed FGFR3 promoter truncation sequences and transfected into ONECUTE2 overexpressed PA cells. We revealed sequence of −2300~−2000 bp had remarkable lower dual luciferase activity, compared to −3000~−2000 bp (Figure 5d), indicating that ONECUT2 transcriptionally regulated FGFR3 expression.

Moreover, NF-CM, TAF-CM, and FGF (ligand for FGFR3 [34]) were applied on FGFR3 knocked-down PA cells (Figure 5e), and we observed that TAF-CM as well as FGF induced tumor growth can be notably attenuated by FGFR3 knockdown (Figure 5f,g). The expression of FGFR3-regulated signaling pathways such as MAPK, mTOR, and PI3K [35] were active by TAF-CM and also decreased after the knockdown of FGFR3 (Figure 5h). In conclusion, ONECUT2 transcriptionally regulated FGFR3 expression in PA, thereby promoting PA progression through MAPK pathway.

### 3.7. CSRM6119 and ABT-263 Combined with Clinical Medications Significantly Inhibited PA Growth

CSRM617 is a small-molecule inhibitor of ONECUT2 that binds directly to the ONECUT2-HOX structural domain. The growth of PA in vivo was inhibited by CSRM617 (Figure 6a). In addition, combined CSRM617 could enhance the efficacy of octreotide (OCT) and significantly inhibit the growth of PA in vivo (Figure 6a). In addition, co-treatment of CSRM617 and OCT attenuated the expression of ki-67 (Figure 6b) and MMP9 (Figure 6c) on tumor tissues, which indicated the significant suppression of tumor cell proliferation and invasion activity.

ABT-263 is a proven inhibitor of fibroblasts [36]. To test the potential pharmacological effects of ABT-263 on PA, ABT-263 combined with octreotide (OCT) or bromocriptine (BRC) was administered to PA-inoculated mice. Tumor volumes were remarkably decreased in the combined treatment group (Appendix A). Moreover, collagen density was significantly decreased after ABT-263 administration determined by Masson staining (Appendix A). Synergistic tumor suppression effect by ABT-263 and somatostatin analog or bromocriptine was observed determined by Ki-67 staining (Appendix A). Using co-staining α-SMA with +TUNEL staining, we also found elevated fibroblast apoptosis in tumors with ABT-263 treatment (Appendix A), while there were no significant changes in fibroblast apoptosis in the sole OCT or BRC groups (Appendix A). Taken together, CSRM617 or ABT-263 could and improve the efficacy of OCT or BRC, which potentially could be translated into clinical application for PA treatment.

## 4. Discussion

TAFs are essential components in the tumor microenvironment [37]. In PA, it has been suggested that TAF-derived cytokines may increase PA invasion [24]. However, the detailed mechanism of the interaction is still poorly understood. Our study indicated that TAF-derived circDennd1b could be transported by exosomes and regulated PA cells to promote tumor progression.

There are a growing number of studies illustrating that circRNAs played a significant role in the tumorigenesis and development of cancer [38]. CeRNA is one of the most common mechanisms [39]. It has been previously reported that circRNAs can act as miRNA sponges that regulate the expression and activity of target genes [40,41,42]. A significant decrease in miR-145-5p was also found in gastric cancer [43] and non-functioning PA [44], In our study, after prediction by bioinformatics analysis and confirmation by luciferase and RNA pull-down experiments, we found that circDennd1b could act as a miRNA sponge for miR-145-5p. Further functional studies showed that circDennd1b could mimic the PA progression-promoting effect of miR-145-5p antagomir and was diminished by miR-145-5p agomir, suggesting that miR-145-5p is a functional target of circDennd1b.

ONECUT2 is a transcription factor related to tumor cell proliferation, angiogenesis, and metastasis [33,45,46]. It has been proved that ONECUT2 regulated the aggressive tumor biology in neuroendocrine prostate cancer through activating SMAD3-HIF1α signaling [31]. Furthermore, ONECUT2 functioned as an oncogene to facilitate HCC metastasis by directly binding to the promoters of fibroblast growth factor 2 (FGF2) and ATP citrate lyase (ACLY) and transcriptionally upregulated their expression [32]. In our research, we found that miR-145-5p can directly bind to ONECUT2. The overexpression of miR-145-5p can significantly reverse the promoting effect of ONECUT2 on the proliferation, migration, and invasion of PA cells.

FGF and its four tyrosine kinase receptors (FGFR1-4) activate multiple downstream signaling pathways. MAPK is the classical downstream pathway of FGF/FGFR [47]. It was previously found that FGFR3 was associated with the occurrence and progression of cancer [34,48]. It has been proved that MAPK signaling promotes cancer progression, such as nasopharyngeal carcinoma [49]. In our previous study, we found that the EGFR/ERK signaling pathway promotes cell migration, invasion, and proliferation in PA [7], suggesting that the MAPK pathway was involved in the PA progression. Our present study proved that ONECUT2 activates downstream MAPK signal pathways by binding downstream FGFR3, thus promoting PA progression.

As a small-molecule inhibitor of ONECUT2, CSRM617 could directly inhibit the expression of ONECUT2. Through in vivo experiments, the growth of GH3 and AtT20 cells was found to be significantly inhibited by CSRM617; further, CSRM617 could improve the efficacy of OCT. Moreover, ABT-263 is a small molecule Bcl-2 inhibitor that can induce apoptosis [50]. Some studies have pointed out that this drug can induce the apoptosis of activated myofibroblasts and reduce the expression of fibrosis genes, thus reducing fibrosis [36,51]. We have proved that ABT-263 can inhibit the fibrosis level of PA, thus further inhibiting the PA growth, which can be translated into the clinical application of PA treatment.

However, we only explored the biological behavior of pituitary adenomas, ignoring the effect of TAFs on hormone secretion. In addition, orthotopic mouse models were not constructed in the study. Besides, a more comprehensive conclusion would be drawn if the extracted primary fibroblasts were subjected to whole transcriptome sequencing.

## 5. Conclusions

In conclusion, our study proved that TAFs promote PA progression by exosomal circDennd1b. circDennd1b, as a ceRNA, upregulated the expression of target gene ONECUT2 by sponging miR-145-5p, thereby transcriptionally regulated FGFR3 and activating downstream MAPK pathway and finally promoting the PA progression. In addition, ABT-263 can improve the efficiency of clinical drugs for PA.

## Figures and Tables

**Figure 1 cancers-15-03375-f001:**
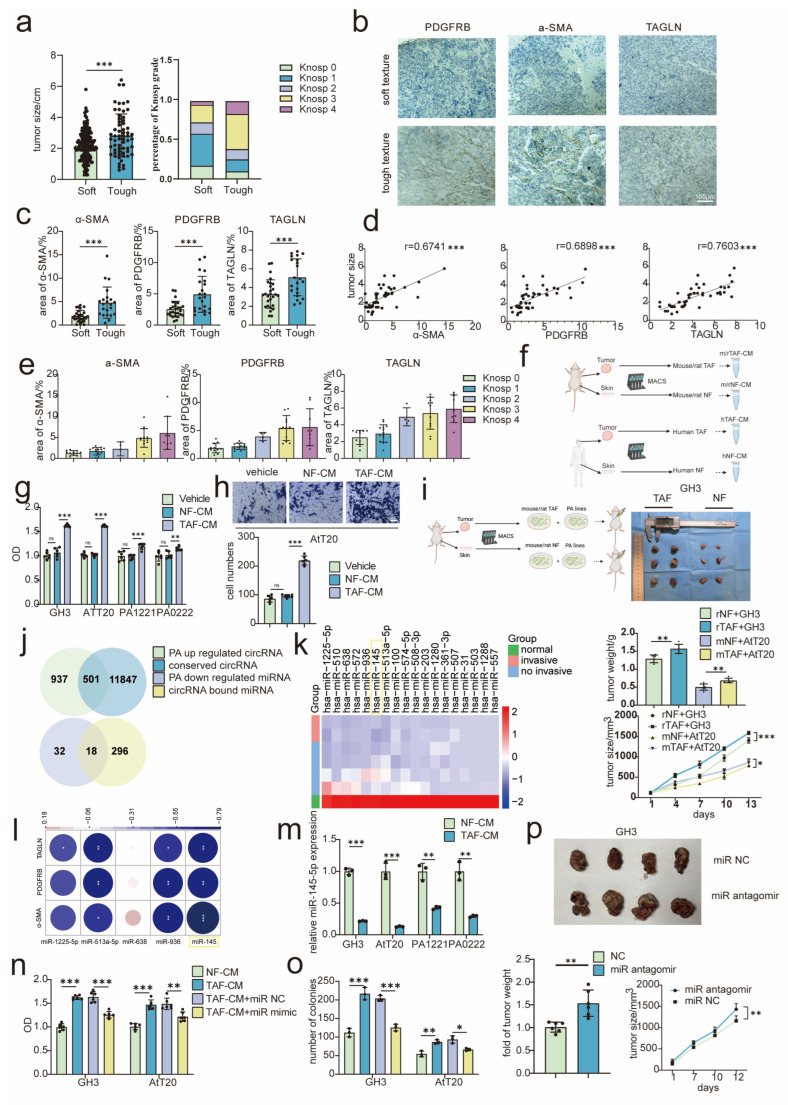
TAFs promoted PA progression by inhibiting miR-145-5p in PA cells. (**a**) Correlation of PA texture with PA size and aggressiveness (n = 242). (**b**) Immunohistochemical staining of TAFs indicators (α-SMA, PDGFRB, TAGLN) between soft and tough textual tumor (scale bar = 100 μm) (n = 49). (**c**–**e**) Correlation of TAFs density with PA size and aggressiveness. (**f**) Schematic diagram of extracting NFs and TAFs (figure created with Biorender.com). (**g**) Cell viability experiments of showing effect of TAFs supernatant on indicated PA cells (n = 6), PA1221 and PA0222 were primary PA cells. (**h**) Transwell assay of effect of TAFs supernatant on indicated PA cells (n = 6). (**i**) Analysis of tumor volume and tumor weight after indicated PA cells were mixed with NFs or TAFs (n = 6) (figure was created with Biorender.com). (**j**) Venn diagram of upregulated circRNAs and downregulated miRNAs in PA. (**k**) Heatmap of downregulated miRNAs in PA. (**l**) Correlation analysis of 5 miRNAs with TAF density (n = 15). (**m**) MiR-145-5p expression level after TAFs supernatant effect on PA cells (n = 3). (**n**) Cell viability experiments of effect of TAFs supernatant on PA cells transfected with miR-145-5p mimic (n = 6). (**o**) Colony formation assay of TAFs supernatant effect on PA cells transfected with miR-145-5p mimic (n = 3). (**p**) Analysis of tumor size and weight after miR-145-5p antagomir injection (5 nmol/3 days) (n = 6) (scale bar = 10 mm). The original magnification was ×200. Data were expressed as mean ± SD. * *p* < 0.05, ** *p* < 0.01, *** *p* < 0.001. Pearson correlation and linear regression were used to compare these parameters between two variables. One-way ANOVA was used for multi-sample comparison and Student’s *t*-test was used for comparison between two groups. Statistical significance was considered to be indicated by a value of *p* < 0.05. CM: conditioned medium.

**Figure 2 cancers-15-03375-f002:**
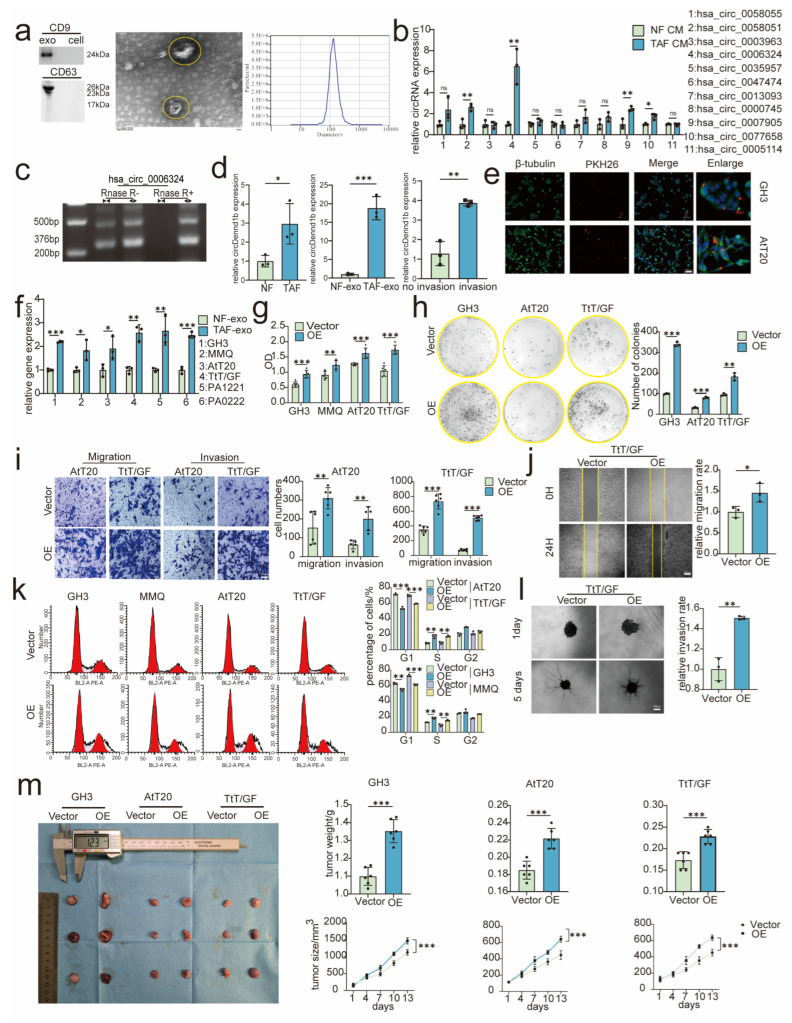
CircDennd1b promoted the proliferation, migration, and invasion of PA cells. (**a**) Exosomes were identified by Western blotting and electron microscopy (scale bar = 100 nm). (**b**) The expression levels of the screened circRNAs in PA tissues (n = 3). (**c**) Verification of circRNA loop formation by RNA electrophoresis (n = 3). (**d**) Expression levels of circDennd1b in TAFs, TAF-exo and aggressive PA (n = 3). (**e**) Interaction between TAF-derived exosomes and PA cells was detected by immunofluorescence (scale bar = 100 μm) (n = 3). (**f**) Expression levels of circDennd1b after TAF-derived exosomes were absorbed by PA cells (n = 3). (**g**–**l**) The effect of circDennd1b on PA cell proliferation was determined by cell viability experiments (n = 6) (**g**), colony formation (n = 3) (**h**), and cell cycle (n = 3) (**k**), and the effect of circDennd1b on PA cell migration and invasion were determined by Transwell assay (n = 6) (**i**) (scale bar = 100 μm),wound healing assay (n = 3) (**j**) (scale bar = 100 μm), and 3D cell invasion (n = 3) (**l**) (scale bar = 100 μm). (**m**) Analysis of tumor size and weight after circDennd1b overexpression (n = 6). The original magnification was ×200. Data were expressed as mean ± SD. * *p* < 0.05, ** *p* < 0.01, *** *p* < 0.001. One-way ANOVA was used for multi-sample comparison and Student’s *t*-test was used for comparison between two groups. Statistical significance was considered to be indicated by a value of *p* < 0.05. circDennd1b OE is abbreviated as OE in Figure 2. exo: exosome.

**Figure 3 cancers-15-03375-f003:**
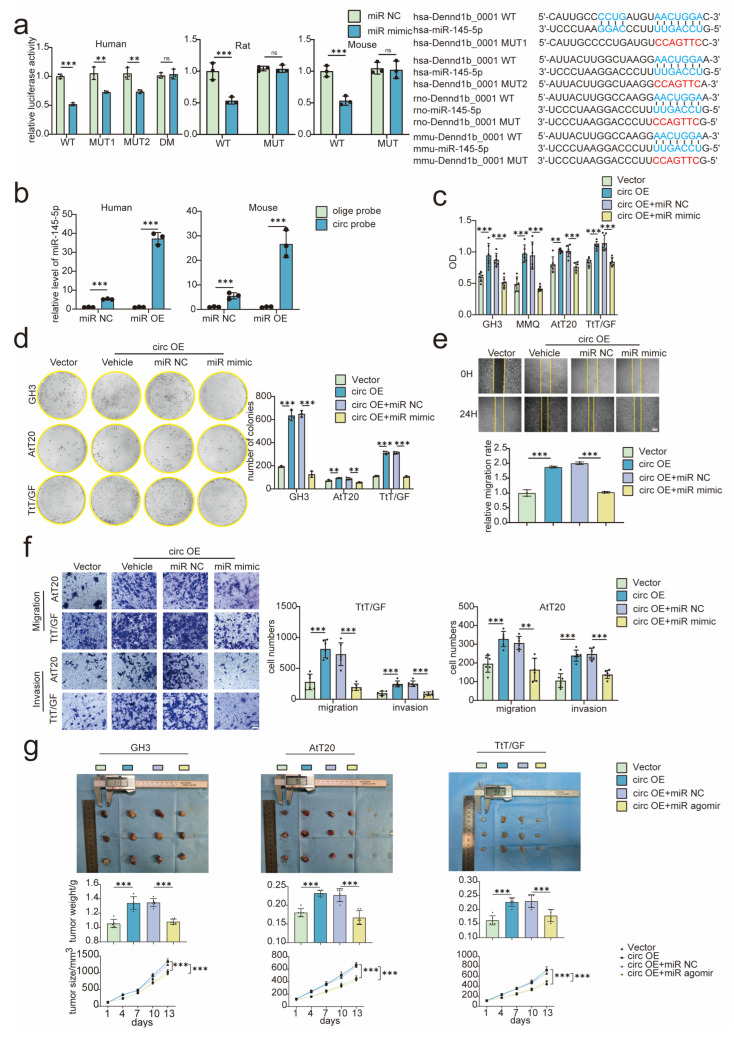
MiR-145-5p reversed the promoting effect of circDennd1b on PA cells. (**a**) After co-transfection with the vector of circDennd1b-WT, MUT, or DM and miR-145-5p mimic or NC, the luciferase activities were detected (n = 3). (**b**) Expression levels of biotin-labeled circDennd1b-bound miR-145-5p (n = 3). (**c**–**f**) PA cells were co-transfected with circDennd1b or Vector, and miR-145-5p mimic or NC: cell proliferation abilities were determined by cell viability experiments (n = 6) (**c**) and colony formation (n = 3), (**d**) cell migration and invasion abilities were determined by Transwell assay (n = 6) (**e**) (scale bar = 100 μm) and wound healing assay (n = 3) (**f**) (scale bar = 100 μm). (**g**) Analysis of tumor size and weight after co-transfected circDennd1b or Vector and miR-145-5p agomir (2 nmol/3 days) or NC (n = 6). The original magnification was ×200. Data were expressed as mean ± SD., ** *p* < 0.01, *** *p* < 0.001. One-way ANOVA was used for multi-sample comparison and Student’s *t*-test was used for comparison between two groups. Statistical significance was considered to be indicated by a value of *p* < 0.05. WT: wild type, MUT: mutation, DM: double mutation. circDennd1b OE is abbreviated as circ OE, and miR-145-5p is abbreviated as miR in the Figure 3.

**Figure 4 cancers-15-03375-f004:**
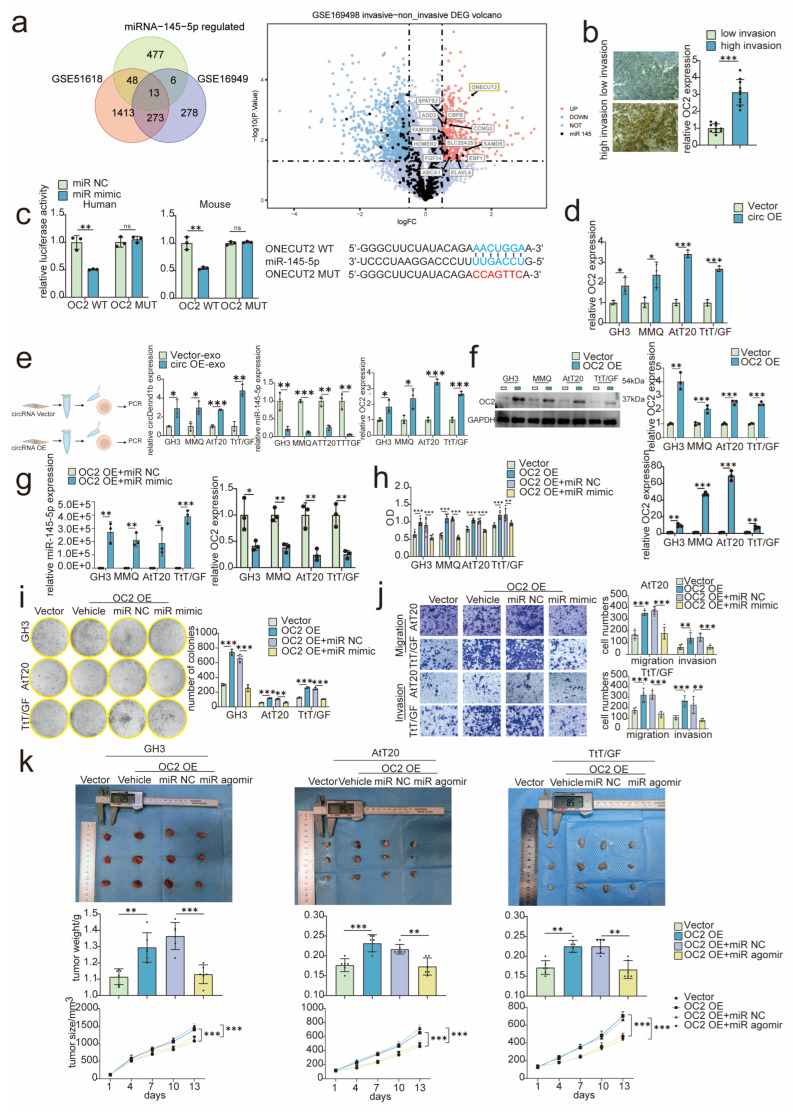
MiR-145-5p reversed the promoting effect of ONECUT2 on pituitary adenoma. (**a**) The potential target gene of miR-145-5p was predicted by Targetscan, GEO. (**b**) The expression level of ONECUT2 in PA tissues was detected by immunohistochemistry (n = 20) (scale bar = 100 μm). (**c**) After co-transfection with the vector of ONECUT2-WT or ONECUT2-MUT and miR-145-5p mimic or NC, the luciferase activities were detected (n = 6). (**d**) The expression level of ONECUT2 after overexpressing circDennd1b by qRT-PCR (n = 3). (**e**) Expression levels of circDennd1b, miR-145-5p, and ONECUT2 in PA by mimicking NF-PA interaction after overexpressing circDennd1b in NFs (n = 3) (figure was created with Biorender.com). (**f**) Expression levels of ONECUT2 by Western blotting and qRT-PCR (n = 3). (**g**) Expression of miR-145-5p and ONECUT2 after co-transfected with ONECUT2 or Vector and miR-145-5p mimic or NC (n = 3). (**h**–**k**) PA cells were co-transfected with ONECUT2 or Vector, and miR-145-5p mimic or NC: cell proliferation abilities were determined by cell viability experiments (n = 6) (**h**) and colony formation assays (n = 3) (**i**), cell migration and invasion abilities were determined by Transwell assay (n = 6) (**j**) (scale bar = 100 μm). (**k**) Analysis of tumor size and weight after co-transfection of ONECUT2 or Vector and miR-145-5p agomir (2 nmol/3 days) or NC (n = 6). The original magnification was ×200. Data were expressed as mean ± SD. * *p* < 0.05, ** *p* < 0.01, *** *p* < 0.001. One-way ANOVA was used for multi-sample comparison and Student’s *t*-test was used for comparison between two groups. Statistical significance was considered to be indicated by a value of *p* < 0.05. OC2: ONECUT2. miR-145-5p was abbreviated as miR in the Figure 4.

**Figure 5 cancers-15-03375-f005:**
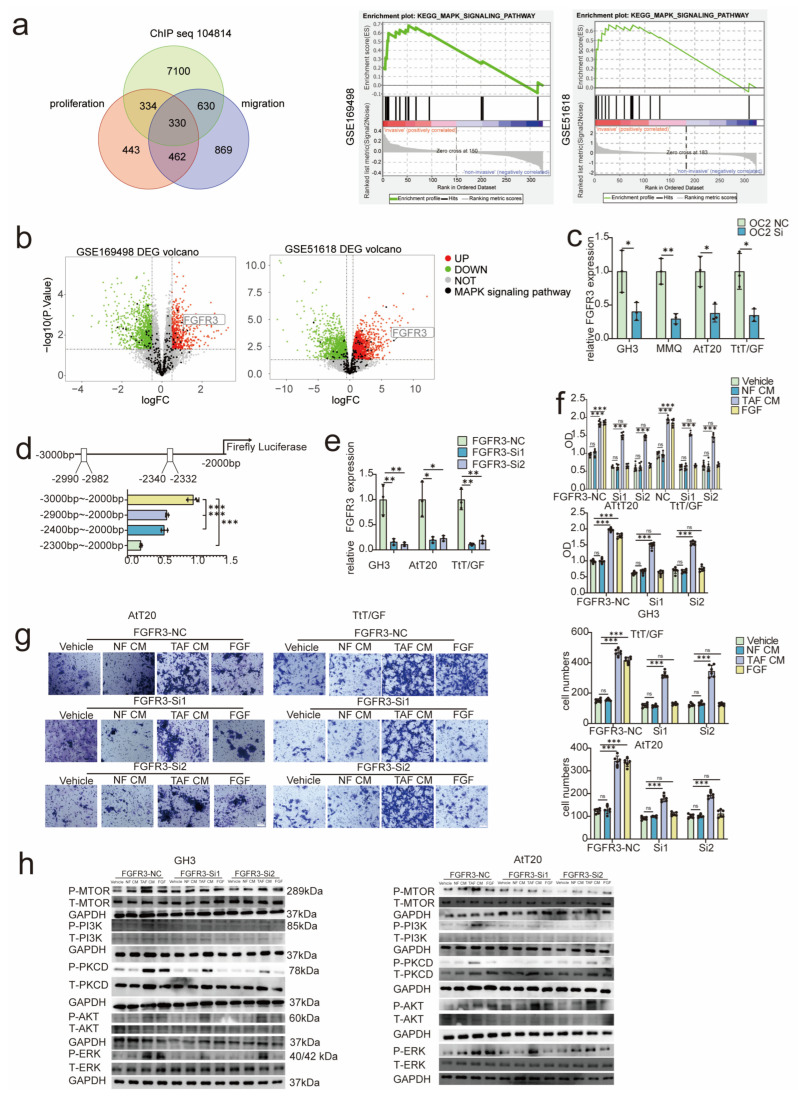
ONECUT2 promoted the expression of FGFR3 and activated the MAPK pathway. (**a**,**b**) The potential target gene of ONECUT2 was visualized by Venn diagram (**a**), pathway enrichment (**a**) and volcano plot (**b**). (**c**) qRT-PCR for FGFR3 expression level after ONECUT2 knockdown (n = 3). (**d**) Dual luciferase activity in promoter regions of different lengths was detected (n = 3). (**e**) qRT-PCR for FGFR3 expression level after FGFR3 knockdown (n = 3). (**f**,**g**) The effect on PA cells after FGFR3 knockdown through cell viability experiments (n = 6) (**f**) and Transwell assay (n = 6) (**g**). (**h**) The expression of FGFR3-regulated signaling pathways after FGFR3 knockdown and NF-CM or TAF-CM activation (n = 3). Data were expressed as mean ± SD. One-way ANOVA was used for multi-sample comparison and Student’s *t*-test was used for comparison between two groups. Statistical significance was considered to be indicated by a value of *p* < 0.05. * *p* < 0.05, ** *p* < 0.01, *** *p* < 0.001.

**Figure 6 cancers-15-03375-f006:**
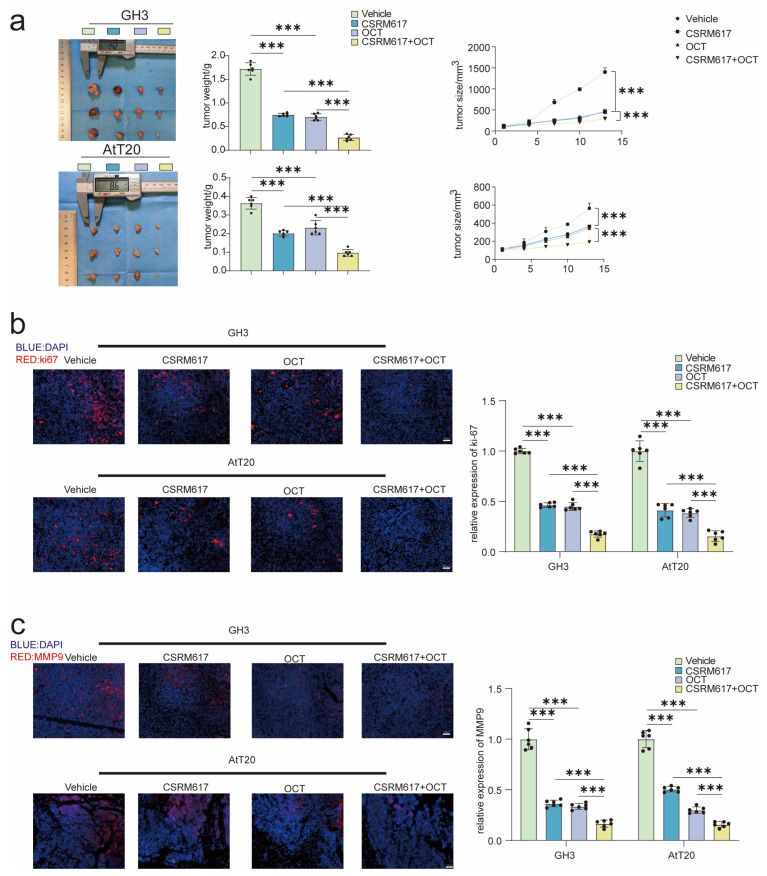
CSRM617 improved efficiency of clinical drugs for PA. (**a**) Analysis of tumor size and weight after the combination of CSRM617 + OCT in nude mice with implants PA cells (n = 6). (**b**) Ki67 staining to detect PA cell activity after combination of CSRM617 + OCT (n = 6) (scale bar = 100 μm). (**c**) MMP9 staining to detect PA cell invasion after combination of CSRM617 + OCT (n = 6) (scale bar = 100 μm). Data were expressed as mean ± SD. One-way ANOVA was used for multi-sample comparison and Student’s *t*-test was used for comparison between two groups. Statistical significance was considered to be indicated by a value of *p* < 0.05. *** *p* < 0.001. OCT: octreotide.

## Data Availability

The datasets generated during and/or analyzed during the current study are available in the GEO (https://www.ncbi.nlm.nih.gov/geo, accessed on 29 January 2022) and CistromeDB (http://cistrome.org/db, accessed on 25 February 2022) repository.

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
