# Peer review of "Tumor-Associated Fibroblast-Derived Exosomal circDennd1b Promotes Pituitary Adenoma Progression by Modulating the miR-145-5p/ONECUT2 Axis and Activating the MAPK Pathway"

_cancers, 2023, doi:10.3390/cancers15133375_

Round 1

Reviewer 1 Report

It is a well thought and designed work.

Author Response

It is a well thought and designed work.

Response:Thank you for your kind approval, while we still further strengthen our evidence of identification of exsomal CircDennd1b and regulation of miR-145-5p by adding additional data and description.  In addition, we have also fixed the typos and revised our figures for better illustration.

Reviewer 2 Report

Jiang et al. reported here that circDennd1b ientified from exosomes of tumor-associated fibroblasts (TAFs) is able to down-regulate miR-145-5p expression in pituitary adenoma (PA) and promote tumor growth. 

1. There are many mistake in manuscript preparation. Take Figure 1 for example, in "Method", the expression of cell number 2×105 is confused. In Figure 1g, the color for TAF-CM label is same as NF-CM, Figure 1f, the meaning of Knosp 0 has not be described in the text. and Figure 1o, the detail of antagomir injection should be provided. The authors should check the manuscript carefully and revised the error. 

2. The retionale for why miR145-5p is focused on this study is very weak. The authors used only one sentance "Among them, miR-145-5p has been proved to be related to tumor growth and invasion in many studies[27-29]" while 18 miRNAs were significantly down-regulated. As key target in this manuscript, more explaination should be provided for choosing miR-145-5p but not other 17 miRNA.  

3. The presentation of heatmap in Figure 1k is not appropriated. Are all the 18 miRNAs expressed in common group with the same level?

4.  Figure 2 and Figure 3 did show clearly the identification of CircDennd1b in exosomes and the regulation of miR-145-5p. 

5. In figure 4, the 13 potential targets for miR-145-5p should be provided, and again, the rationales for choosing ONECUT2 should be addressed more. 

6. It is not surprised that ABT-263, an inhibitor of fibroblasts, can improve efficiency of clinical drugs for PA, since the role of TAFs in promote PA have been demonstrated in many studied. The use of ABT-263 can not strenthen the importance of circDennd1b 2 to modulate miR-145-5p/ONECUT2 axis in PA.

Moderate editing of English language is needed.

Author Response

1.There are many mistake in manuscript preparation. Take Figure 1 for example, in "Method", the expression of cell number 2×105 is confused. In Figure 1g, the color for TAF-CM label is same as NF-CM, Figure 1f, the meaning of Knosp 0 has not be described in the text. and Figure 1o, the detail of antagomir injection should be provided. The authors should check the manuscript carefully and revised the error. 

Response: Thank you for your comments, we apologize for the mistakes. As suggested, we have corrected the errors in the manuscript, fixed the error on cell numbers (actually it’s 105), and we have revised the colors of the labels in Fig.1g. The meaning of Knosp grade in Fig.1 was also now described in the main text, and the injection details in Fig.1p can be found in the methods. In addition, we have also revised some of the typos.

  1. The retionale for why miR145-5p is focused on this study is very weak. The authors used only one sentance "Among them, miR-145-5p has been proved to be related to tumor growth and invasion in many studies[27-29]" while 18 miRNAs were significantly down-regulated. As key target in this manuscript, more explaination should be provided for choosing miR-145-5p but not other 17 miRNA.  

Response: Thank you for your comments. Our hypothesis of this study is that TAFs derived circRNAs could regulate miRNA/mRNA according to the ceRNA conception. While there is no exist evidence to illustrate this in pituitary adenoma (PA) study. Therefore, taking advantage of published circRNA-seq data in PA, we first identified the up-regulated circRNAs (1438 (937+501) up-regulated, Fig 1j) in PA comparing to normal pituitary tissues. And since the experiment setting on PA would be performed on rat or murine originated tissues (GH3, AtT20, MMQ, TtT/GF as well as in vivo models), we further identified the conserved up-regulated circRNAs (501, Fig 1j). Next, the ceRNA network was established using circinteractome (https://circinteractome.nia.nih.gov/). There are only 18 miRNAs that are down-regulated in PA and can be potentially bind to identified circRNAs (Fig 1j and Fig 1k). Furthermore, we compared the 18 miRNAs in invasive and non-invasive PA (GSE46294) and the top-5 most notable downregulated (according to the fold change) miRNAs were chosen because the target miRNA should also be associated with invasive. To further identify potential targets, since our hypothesis is that TAFs regulates the miRNA expression in PA, the correlation analysis was then performed between these down-regulated miRNAs expression level and TAFs density in 15 pituitary adenoma tissues (Fig 1l). We revealed that miR-145 was the one that most significant negatively correlated with TAFs density. Therefore, we finally chose miR-145 as our target miRNA. We have now added the description of target identification in the text and the correlation analysis figure in the Fig 1l.

  1. The presentation of heatmap in Figure 1k is not appropriated. Are all the 18 miRNAs expressed in common group with the same level?

Response: Sorry we didn't describe it clearly, in this transcriptome sequencing data, the miRNA expression in normal pituitary has been normalized to 1 and the miRNA expression was presented as fold change. It is expected that the values are 1 for GSM1128224 (normal tissue) and GSM1128228 (normal tissue). This is the data that we retrieved from GEO database (GSE46294).  We have now added the description in the text for easier understanding.

  1. Figure 2 and Figure 3 did show clearly the identification of CircDennd1b in exosomes and the regulation of miR-145-5p. 

Response: Thank you for your kind approval, while we still further strengthen our evidence of identification of exsomal CircDennd1b and regulation of miR-145-5p by adding additional data and description.

As we newly added description in the text and the answer from question #2, we first identified the up-regulated circRNAs (1438 (937+501) up-regulated, Fig 1j) in PA comparing to normal pituitary tissues. And since the experiment setting on PA would be performed on rat or murine originated tissues (GH3, AtT20, MMQ, TtT/GF as well as in vivo models), we further identified the conserved up-regulated circRNAs (501, Fig 1j). Next, the ceRNA network was established using circinteractome (https://circinteractome.nia.nih.gov/). There are only 18 miRNAs that are down-regulated in PA and can be potentially bind to identified circRNAs (Fig 1j and Fig 1k). Furthermore, we compared the 18 miRNAs in invasive and non-invasive PA (GSE46294) and the top-5 most notable downregulated (according to the fold change) miRNAs were chosen because the target miRNA should also be associated with invasive. To further identify potential targets, since our hypothesis is that TAFs regulates the miRNA expression in PA, the correlation analysis was then performed between these down-regulated miRNAs expression level and TAFs density in 15 pituitary adenoma tissues (Fig 1l). We revealed that miR-145 was the one that most significant negatively correlated with TAFs density. Therefore, we finally chose miR-145 as our target miRNA.

        To further identify potential circRNAs that regulate miR-145-5p, we retrieved the analysis data from our established circRNA/miRNA ceRNA network (we have now added this axis in the Table S3)(Fig S2), and we revealed that 11 circRNAs that containing multiple binding sites (more than 1) with miR-145-5p. Next, to verify that the potential circRNA role in TAF derived exosomes, we first treated primary pituitary adenoma cells with normal fibroblast conditioned medium (NF CM) or TAF CM, and the expression level of these 11 circRNAs were determined by RT-qPCR. As shown in Fig 2b, hsa_circ_0006324 (also named circDennd1b) was the most up-regulated one. Furthermore, we have now added the data of exosomal circDennd1b in NF or TAF (Fig. 2d). As expected, both cellular and exosomal circDennd1b was remarkable elevated in TAF (Fig. 2d).

Regarding the regulation role of circDennd1b on miR-145-5p, the dual luciferase assay, RNA pull-down and functional assays (CCK8, colony formation, wound healing, transwell as well as in vivo models) were performed to comprehensively validate the direct regulation axis between circDennd1b and miR-145-5p in PA (Fig. 3a-g).

  1. In figure 4, the 13 potential targets for miR-145-5p should be provided, and again, the rationales for choosing ONECUT2 should be addressed more. 

Response: Thank you for the suggestion. As suggested by the reviewer, we have now labeled all 13 potential target genes regulated by miR-145-5p in the volcano plot (Fig.4a). ONECUT2 was the most significantly up-regulated one among them in the invasive PA compared to non-invasive PA using RNA-seq datasets from GSE169498. Therefore ONECUT2 was selected as the target gene of miR-145-5p in PA. We have now added this description in the main text.

  1. It is not surprised that ABT-263, an inhibitor of fibroblasts, can improve efficiency of clinical drugs for PA, since the role of TAFs in promote PA have been demonstrated in many studied. The use of ABT-263 can not strenthen the importance of circDennd1b 2 to modulate miR-145-5p/ONECUT2 axis in PA.

Response: Thank you for your suggestion. Our hypothesis is that TAF derived exosomal circDennd1b regulates miR-145/ONECUT2 axis in PA, therefore we have now performed the in vivo experiments using ONECUT2 inhibitor CSRM617 in treating PA murine models (newly generated Fig. 6). The ONECUT2 inhibitor CSRM617 was combined with and current medication for PA. As shown in our newly generated figure 6, we observed that CSRM617 remarkably reduced tumor size and improved the efficacy of octreotide, validating the role of circDennd1b/miR-145-5p/ONECUT2 axis on pituitary adenoma, and ONECUT2 inhibitor CSRM617 might be a potential options for treating pituitary adenoma while further trials would be needed.

7.Moderate editing of English language is needed.

Response: We have now substantially improved the manuscript and revised the language  as suggested.

Reviewer 3 Report

In this study, Jiang et al., analyzed the role of tumor-associated fibroblasts (TAF) in pituitary adenoma progression. They found that conditioned media of TAF increased migration and invasion of PA cells. As the underlying mechanism they described that fibroblast-derived exosomal circDennd1b decreased miR-145-5p expression followed by enhanced production of the target gene ONECUT2 favoring MAPK pathway activation. At this stage, the data of the manuscript are interesting however, some questions remain before the manuscript can be accepted for publication.

Specific comments:

-        Figure 1: explanation of figure 1a (classifications of knosp) is missing in the text and material and methods.

-        In situ stainings of fibroblasts markers: brown stainings are hard to detect. Evaluation of percentages of TAF is missing. How was the TAF content determined? Please include a score.

-        Figure 2: in 2a western blot of CD9 is to dark and the results are hard to detect.

-        Figure 4b: quantification of OC2 in situ stainings: how many biopsies were analyzed? Figure legends described n=6, but quantifications showed more dots. Please specify and explain evaluation of relative OC2 expression in high and low invasive tumors.

-        Figure 4e: the authors showed enhanced OC2 expression after circDennd1b over-expression. What represents the third figure in e (Y-Axis is not labeled)? What about miR-142-5p expression in this experimental setup?

-        Figure 5: the authors identified FGFR3 as an ONECUT2 regulated gene. Data of FGFR3 expression in patient samples are missing. Is there a correlation between ONECUT2 and FGFR3 expression in situ?

-        Figure 6: the authors used the ABT-263 inhibitor to block fibroblasts. Please specify usage of this inhibitor. ABT-263 is an inhibitor for fibroblasts, but not related to the defined mechanism. MAPK inhibitors are available.

Further comments:

The authors claim reduced miR-142-5p expression in PA tissues. Data regarding miR-142-5p expression in tumor tissues versus healthy tissues are missing. A statement concerning miR-142-5p expression is missing in the discussion with regard to enhanced miR-142-5p expression in other cancers.

Pictures of the colony formation assays are to dark/to blue to enable detection of colonies.

A statistic section is missing. The authors should include how many experiments have been done and used for data presentation for each figure, statistical analysis and explain it in the corresponding legends. How many biological or technical replicates? What statistical test for significances?

The authors defined a complex axis in cell cultures and mouse experiments but validation of some data in situ or patients samples are missing and should be provided.

Author Response

Specific comments:

-        Figure 1: explanation of figure 1a (classifications of knosp) is missing in the text and material and methods.

Response: Thanks for the comments. The description of Knosp grade in Fig.1a has now been added in the text.

-        In situ stainings of fibroblasts markers: brown stainings are hard to detect. Evaluation of percentages of TAF is missing. How was the TAF content determined? Please include a score.

Response: According to previous studies, markers including α-SMA, PDGFRB, and TAGLN have long been considered as markers of TAFs. Therefore, the density of TAFs in PA were determined by detecting positive area of α-SMA, PDGFRB, or TAGLN relative to area of whole image using ImageJ software (Fig. 1b-e). We have now added the method details in the Materials & Methods part.

-        Figure 2: in 2a western blot of CD9 is to dark and the results are hard to detect.

Response: Thanks for the comments. We re-examined the CD9 expression in exosomes and presented for better illustration.

-        Figure 4b: quantification of OC2 in situ stainings: how many biopsies were analyzed? Figure legends described n=6, but quantifications showed more dots. Please specify and explain evaluation of relative OC2 expression in high and low invasive tumors.

      Response: We are sorry for the erroneous sample size described in figure legends. in detecting ONECUT2 expression, we analyzed the expression levels in 10 low (Knosp 0-2) and 10 high (Knosp 3-4) invasive pituitary adenomas, respectively. ONECUT2 expression level in the low invasion group compared to the high invasion group was 12.00±3.104 vs 37.56±9.042 (p<0.001). We have revised the description in the figure legends and added more details in the Materials & Methods part.

-        Figure 4e: the authors showed enhanced OC2 expression after circDennd1b over-expression. What represents the third figure in e (Y-Axis is not labeled)? What about miR-142-5p expression in this experimental setup?

      Response: Thanks for the comments. We have added labeling of Y-axis in the Fig 4e for all the individual figures as suggested to clarify actual Y-axis.  Also, in Fig.4e, what we performed here are as follow: fibroblast was overexpressed with circDennd1b and exosomes were subsequently harvested to treat PA cells. As suggested, we have added the data of miR-145-5p expression level in PA cells after treated with exosomes from circDennd1b overexpressed fibroblast (Fig. 4e), and as expected, exosome derived from circDennd1b overexpressed fibroblast inhibits the miR-145-5p expression in PA. We have now revised the figure labeling and figure legends for better understanding.

-        Figure 5: the authors identified FGFR3 as an ONECUT2 regulated gene. Data of FGFR3 expression in patient samples are missing. Is there a correlation between ONECUT2 and FGFR3 expression in situ?

Response: As suggested, we have now added the immunohistochemical staining of FGFR3. We observed that the FGFR3 expression level was higher in the invasive PA group. In addition, the FGFR3 expression was positively correlated with the ONECUT2 expression level in patients’ tissues (Fig. S7).

-        Figure 6: the authors used the ABT-263 inhibitor to block fibroblasts. Please specify usage of this inhibitor. ABT-263 is an inhibitor for fibroblasts, but not related to the defined mechanism. MAPK inhibitors are available.

Response: Regarding the usage of ABT-263, we have described in the method, 75mg/kg every two days by intraperitoneal injection. Besides, our hypothesis is that TAF derived exosomal circDennd1b regulates miR-145/ONECUT2 axis in PA, therefore we have now performed the in vivo experiments using ONECUT2 inhibitor CSRM617 in treating PA murine models. The ONECUT2 inhibitor CSRM617 was combined with and current medication for PA. As shown in our newly generated figure 6, we observed that CSRM617 remarkably reduced tumor size and improved the efficacy of octreotide, validating the role of circDennd1b/miR-145-5p/ONECUT2 axis on pituitary adenoma, and ONECUT2 inhibitor CSRM617 might be a potential options for treating pituitary adenoma while further trials would be needed.

Further comments:

The authors claim reduced miR-142-5p expression in PA tissues. Data regarding miR-142-5p expression in tumor tissues versus healthy tissues are missing. A statement concerning miR-142-5p expression is missing in the discussion with regard to enhanced miR-142-5p expression in other cancers.

Response: According to the results of transcriptome sequencing (GSE46294), we revealed that the expression level of miR-145-5p is significantly reduced in tumor tissues compared to normal pituitary tissues (Fig. 1k). Since our hypothesis is that TAFs regulates the miRNA expression in PA, the correlation analysis was then performed between the down-regulated miRNAs expression level and TAFs density in 15 pituitary adenoma tissues (Fig. 1l). We revealed that miR-145 was the one that most significant negatively correlated with TAFs density. In addition, we also detected that the miR-145-5p level in tumor cells is significantly reduced after the effect of TAF supernatant (Fig. 1m). Besides, we also added a discussion of miR-145-5p with regard to other type of cancers (down-regulated level in gastric cancer, non-function pituitary adenoma) in the discussion.

Pictures of the colony formation assays are to dark/to blue to enable detection of colonies.

Response: Thank you for your suggestion. We readjusted the image of colony formation for better illustration.

A statistic section is missing. The authors should include how many experiments have been done and used for data presentation for each figure, statistical analysis and explain it in the corresponding legends. How many biological or technical replicates? What statistical test for significances?

Response: Thank you for your comments. We have now added the statistical section in the text and the corresponding details in the figure legends.

The authors defined a complex axis in cell cultures and mouse experiments but validation of some data in situ or patients samples are missing and should be provided.

Response: We detected the expression levels of circDennd1b (Fig.2b, Fig. 2d), miR-145-5p(Fig. 1l, newly generated), ONECUT2(Fig. 4b) in patients samples. As suggested by the reviewer, we have now also assessed the FGFR3(Fig. S7) in tumor tissues. Besides, primary pituitary adenoma cells (PA1221, PA0222) from patients were also used in the study to verify the functional axis in regulating corresponding expressions (Fig.1g, Fig. 1m, Fig. 2b, Fig. 2f).

Round 2

Reviewer 2 Report

The authors have addressed comments and I am satisfied with the improvements.

Reviewer 3 Report

The authors adequately answered to the questions. The manuscript has significantly improved.